# Numerical Simulation of Flame Retardant Polymers Using a Combined Eulerian–Lagrangian Finite Element Formulation

**Julio Marti** [1,*,†] **, Jimena de la Vega** [2,†]**, De-Yi Wang** [2,†] **and Eugenio Oñate** [1,†]

1 Centre Internacional de Mètodes Numèrics en Enginyeria (CIMNE), Gran Capitán s/n, 08034 Barcelona, Spain; onate@cimne.upc.edu
2 IMDEA Materials Institute, C/Eric Kandel, 2, Getafe, 28906 Madrid, Spain; jimena.vega@imdea.org (J.d.l.V.); deyi.wang@imdea.org (D.-Y.W.)
* Correspondence: julio.marti@cimne.upc.edu
† These authors contributed equally to this work.

**Abstract:** Many polymer-made objects show a trend of melting and dripping in fire, a behavior that may be modified by adding flame retardants (FRs). These affect materials properties, e.g., heat absorption and viscosity. In this paper, the effect of a flame retardant on the fire behavior of polymers in the UL 94 scenario is studied. This goal is achieved essentially by applying a new computational strategy that combines the particle finite element method for the polymer with an Eulerian formulation for air. The sample selected is a polypropylene (PP) with magnesium hydroxide at 30 wt.%. For modelling, values of density, conductivity, specific heat, viscosity, and Arrhenius coefficients are obtained from different literature sources, and experimental characterization is performed. However, to alleviate the missing viscosity at a high temperature, three viscosity curves are introduced on the basis of the viscosity curve provided by NIST and the images of the test. In the experiment, we burn the specimen under the UL 94 condition, recording the process and measuring the temperature evolution by means of three thermocouples. The UL 94 test is solved, validating the methodology and quantifying the effect of FR on the dripping behavior. The numerical results prove that well-adjusted viscosity is crucial to achieving good agreement between the experimental and numerical results in terms of the shape of the polymer and the temperature evolution inside the polymer.

**Keywords:** dripping; melt flow; UL 94 test; particle finite element method (PFEM); flame retardant

## 1. Introduction

Once a fire starts in the living area of a residential building, mattresses and upholstered furniture made of polymeric materials are often the first items to enhance its growth. Many of these materials tend to liquefy, flow, and drip, forming flammable pools, which then crucially spread the fire to objects that were originally non-adjacent. This burning behavior influences the onset of the fire, and the flames spread within it.

In order to considerably reduce the risk of these accidents, the introduction of fire retardants (FRs) in the polymer matrix was a key step in inhibiting combustion and smoke generation. This is due to the fact that they affect the fire and material properties in numerous ways [1–4], e.g., heat absorption and melt viscosity.

To assess the flammability of "new materials", there exists a set of small-scale reaction-to-fire tests. One of these is the vertical UL 94 test [5]. In this test, a small-sized specimen is exposed to a flame for 10 s, thereby forcing its ignition. After this period of time, the flame is removed and the burning becomes free.

Apart from fire tests, numerical simulation is another way to assess flammability. In [6], the authors investigate the effect of FRs on the fire behavior of a polycarbonate/acrylonitrile butadiene styrene (PC/ABS) specimen using a Lagrangian finite element model known as the particle finite element method (PFEM). The computational solution procedure of the

UL 94 test using PFEM consists of a non-linear loop, where at each step, the Navier–Stokes equations are solved in conjunction with the temperature equation. Furthermore, the authors of [7] simulate the burning of a polypropylene (PP) with other FRs to understand the complex burning behavior introduced by them. Although in both papers the PFEM was able to simulate the melt flow and dripping, facilitating an understanding of the complex behavior of polymeric materials during fire, none of the works modeled the effect of the surrounding air, and, therefore, a strongly simplified combustion model was used. Furthermore, there was no comparison between the experimental and numerical results in terms of evolution of the free surface of the polymer, temperature inside the polymer, etc. On the other hand, recently, researchers have developed a numerical strategy [8] that can reproduce the UL 94 test much more accurately than other cited models can [6,7,9]. The main difference between the new approach and the previous one is that the new one combines the PFEM for the polymer with an Eulerian formulation for the surrounding air. The coupling is performed using an embedded Dirichlet–Neumann scheme [8]. Although the numerical tool was observed to be a promising methodology for predicting behavior, as the numerical results agree well with the experimental ones corresponding to pure PP, the tool requires reliable input parameters for modelling the behavior of the material in fire situations.

The work proposed in the current study focuses on using the numerical tool developed by the authors to assess the behavior of an FR polymer in the UL 94 set-up. It should be noted that the tool has not previously been used for FRs. Traditionally halogenated FRs are highly effective in the flame retardancy of polymers [10]. However, the application of halogenated FRs has caused an increasing number of environmental problems due to their toxicity and/or bioaccumulation. Therefore, the development and use of halogen-free FRs on polymers has aroused a considerable amount of interest in this field. Amongst the halogen-free FRs, magnesium hydroxide (MDH, $Mg(OH)_2$) is one of the widest applied on polymers due to its low cost and environmental friendliness. In the fire condition, MDH decomposes to magnesium oxide and releases water, which is a typical endothermic reaction [11]. The endothermic decomposition of MDH absorbs some of the heat during the combustion process, delaying both the ignition and the combustion of the polymer. A mixture of magnesium hydroxide at 30 wt.% (MDH30) with polypropylene (PP) is modeled numerically. An experimental characterization is performed to first obtain the main kinetic parameters, such as the Arrhenius coefficient and activation energy. Moreover, the viscosity as a function of the temperature is obtained by performing a rheology study. However, given the fact that the values are still missing for high temperatures, in this work, these are estimated on the basis of the viscosity curve provided by NIST [12] and the images of the experiment for those ranges where the curve is not available. Thus, three viscosity curves are introduced. To validate the numerical results, the polymer is burned in the UL 94 test enriched with three thermocouples inside during the condensate phase. The numerical results are analyzed and compared to explain the dripping behavior during burning, thereby improving the scientific understanding of the impact of MDH on the behavior of the PP polymer.

This paper is structured as follows: first, the system of governing equations describing the behavior of the polymer in fire situations is presented; next, the solver for the polymer and air are introduced, and the overall solution algorithm is outlined; the paper concludes with the simulation of the polymer in the UL 94 set-up.

## 2. Governing Equations

The computer simulation of a polymer in the UL 94 scenario is an extremely complex process involving many phenomena, such as fluid flow, heat transfer, material degradation, and flame chemistry. Such simulation requires two different kinds of computational modules: one that deals with the solid phase (i.e., the polymer itself) to calculate the polymer motion, the thermal degradation, and the volatile release (pyrolysis), and a second one that models the combustion process in the flame formed by the mixing of volatiles in the

surrounding air. The processes in the polymer (solid) and the gas phase are interdependent, and, thus, the two computational modules must be robustly and efficiently coupled. Next, the mathematical formulation describing the phenomena involved during the burning of a polymer in a fire is presented.

Let $\Omega \subset R^3$ be a bounded domain containing the air $\Omega_a$ and the polymer $\Omega_p$ (Figure 1). Note that they are treated as viscous fluids (this assumption becomes satisfactory for higher temperatures and regarding a narrow range of shear rates such as those experienced in the UL 94 test [12]). The problem presented above is governed by the following three-dimensional and unsteady equations written in a compact form as follows:

$$\frac{\partial \rho \phi}{\partial t} + \nabla_x \cdot (\rho \mathbf{v} \phi) = \nabla_x \cdot (H_\phi \nabla_x \phi) + S_\phi \quad \text{in } \Omega \times (0, t) \tag{1}$$

where the variables $\phi$, $H_\phi$ and $S_\phi$ are defined as follows:

| Transport of | $\phi$ | $H_\phi$ | $S_\phi$ | |
|:---:|:---:|:---:|:---:|:---:|
| Mass | 1 | 0 | $\epsilon_v$ | (2) |
| Momentum | $\mathbf{v}$ | $\mu$ | $-\nabla_x p + \mu \nabla_x (\nabla_x^T \mathbf{v}) + \rho \mathbf{f}$ | (3) |
| Energy | $T$ | $\kappa/C$ | $\gamma[w_T/C + (\nabla.Q_R)/C] + (1-\gamma)Q_v/C$ | (4) |
| Species | $Y_k$ | $\kappa/C$ | $-w_k/C \text{ for } k = F \text{ and } O$ | (5) |

with

| Symbol | Parameter |
|:---:|:---:|
| $\nabla_x = \{\partial_{x_i}\}_{i=1}^3$ | vectorial operator of spatial derivatives |
| $\rho$ | density |
| $p$ | pressure |
| $\mu$ | viscosity |
| $\mathbf{f}$ | gravity force |
| $C$ | capacity |
| $\kappa$ | thermal conductivity |
| $w_T$ | rate of production of heat [1] |
| $Q_R$ | radiative heat flux |
| $\epsilon_v$ | mass loss |
| $Q_v$ | heat absorbed due to pyrolysis |
| $A$ | pre-exponential function |
| $E$ | activation energy |
| $R$ | universal gas constant |
| $\alpha$ | absorption coefficient |
| $\sigma$ | Stefan–Boltzmann constant |

[1] Assuming constant values for the Schmidt ($Sc = 1$) and Prandtl ($Pr = 1$) number-simplified composition and temperature-dependent transport properties; thus, $\rho D = \kappa/C$.

The parameters presented in the table above are considered constant, except for the viscosity of the polymer, which is assumed to be a function of temperature. This is defined in Section 4. The parameter $\gamma$ is set to 0 in $\Omega_p$ and to 1 in $\Omega_a$. The following table presents the source terms:

| Source Terms | $S_\phi$ | |
|:---:|:---:|:---:|
| $\nabla \cdot QR$ | $\alpha\left(4\sigma T^4 - G\right)$ | (6) |
| $\epsilon_v$ | $-Ae^{-E/RT}$ | (7) |
| $Q_v$ | $\rho H \epsilon_v$ | (8) |
| $w_F$ | $-CB_c\rho^2 Y_F Y_O \exp^{(-T_a/T)}$ | (9) |
| $w_O$ | $-sw_{C_3H_8}$ | (10) |
| $w_T$ | $h_{C_3H_6}B_c\rho^2 Y_F Y_O \exp^{(-T_a/T)}$ | (11) |

The incident radiation G adopting the P1 method [13] is computed by solving

$$-\nabla \cdot \left(\frac{1}{3\alpha}\nabla G\right) + \alpha G = 4\alpha\sigma T^4 \tag{12}$$

which is subject to the Marchak boundary condition [13].

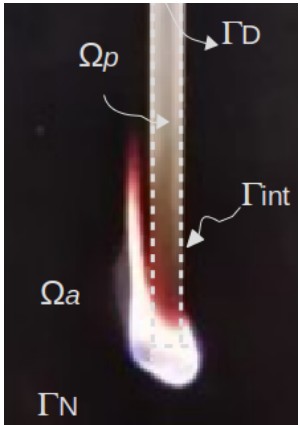

**Figure 1.** Domain.

Note that the pyrolysis model used in this work (see Equation (7)) is based on simple, thermally activated, single-step decomposition. Its validation is presented in [12]. Although it is a simple model, the numerical results presented in [6,7,12,14–17] have revealed that it is good enough to model the pyrolysis during burning.

In order to solve Equation (1), on the external boundaries $\Gamma_D$ and $\Gamma_N$, velocities and tractions are prescribed, respectively, while standard coupling conditions are applied to the internal interfaces $\Gamma_{int}$. For further details on the boundary conditions applied, the reader is referred to [8].

## 3. Numerical Strategy

In order to obtained a numerical solution of the problem presented above, a partitioned strategy is developed. It uses the particle finite element method (PFEM) [18] for the polymer, while a Eulerian formulation is used for the air. To correctly represent the interaction between them, an embedded Dirichlet–Neumann scheme is used. For further details on the numerical strategy, the reader is referred to [8,19]. Next, the discrete version of the problem in time and space and the solution procedure are presented.

### 3.1. Particle Finite Element Method for the Polymer

The particle finite element method (PFEM) [18] is a Lagrangian finite element model. The governing equations are solved on a mesh following the standard FEM methodology. Once the discrete equations are solved, the FEM nodes, which are treated as particles, can move according to their velocity. As a consequence of the node motions, the mesh must be regenerated using Delaunay triangulation [20]. For further details on the PFEM approach,

the reader is referred to [18]. Next, the governing equations together with discrete version of them and the overall algorithm are presented.

The discrete version of the governing equations defined by Equations (13)–(15) is obtained by applying the backward Euler method to the Galerkin variational form of Equations (2)–(4). The unknown velocity, pressure, and temperature at $t^{n+1} = t^n + \Delta t$ can be computed as

$$(\rho\mathbf{M} + \Delta t\mu\mathbf{K})\mathbf{v}^{n+1} = \rho\mathbf{M}\mathbf{v}^n + \Delta t\mathbf{G}\mathbf{p}^n + \Delta t\rho\mathbf{F} \tag{13}$$

$$\mathbf{M}\mathbf{p}^{n+1} = \mathbf{M}\mathbf{p}^n - \Delta t\mathcal{K}\mathbf{D}\mathbf{v}^{n+1} + \Delta t\mathcal{K}\mathbf{F}\epsilon_v \tag{14}$$

$$(\rho C\mathbf{M} + \Delta t\kappa\mathbf{K})\mathbf{T}^{n+1} = \rho C\mathbf{M}\mathbf{T}^n + \mathbf{M}Q_v \tag{15}$$

Note that in the present work, a nearly-incompressible behavior [21] is applied to decouple the pressure from the velocity. The matrices and vectors presented above can be found in [8].

### 3.2. Finite Element Formulation for the Air

Applying a standard Eulerian FEM procedure with a backward Euler scheme [22,23] to the Equations (2)–(12) leads to the discrete form of the governing equations Equations (16)–(20). The unknown incident radiation, temperature, species, velocity, and pressure at $t^{n+1}$ can be computed as

$$(\mathbf{L}_{1/3\alpha} + \mathbf{M}_\alpha)\mathbf{G} = 4\alpha\sigma\mathbf{M}_\alpha\mathbf{T}^{n+1,4} \tag{16}$$

$$\left(C\mathbf{M} + \Delta tC\mathbf{C}(\mathbf{v}^n) + \Delta t\mathbf{L}_{\kappa/\rho}\right)\mathbf{T}^{n+1} = C\mathbf{M}\mathbf{T}^n$$

$$\Delta t\mathbf{M}_{1/\rho}w_T + 4\alpha\sigma\Delta t\mathbf{M}_{1/\rho}\mathbf{T}^{n+1,4} - \Delta t\mathbf{M}_{1/\rho}\mathbf{G} \tag{17}$$

$$\left(\mathbf{M} + \Delta tC\mathbf{C}(\mathbf{v}^n) + \Delta t\mathbf{L}_{\kappa/\rho}\right)\mathbf{Y}_k^{n+1} = C\mathbf{M}\mathbf{Y}_k^n + \Delta t\mathbf{M}_{/\rho}w_k \tag{18}$$

$$(\mathbf{M} + \Delta t\mathbf{C}(\mathbf{v}) + \Delta t\mu\mathbf{K})\mathbf{v}^{n+1} = \mathbf{M}\mathbf{v}^n + \Delta t\mathbf{G}_{1/\rho}\mathbf{p}^{n+1} + \Delta t\mathbf{F} \tag{19}$$

$$\mathbf{D}\mathbf{v}^{n+1} = \mathbf{0} \tag{20}$$

Velocity and pressure from Equations (19) and (20) are uncoupled using the fractional step procedure [24–26]. To stabilize Equations (17)–(20) in space, the algebraic sub-grid scale (ASGS) technique [27] is used.

Having presented the discrete versions of the two domains to solve, the next component to be defined is the overall solution strategy employed.

### 3.3. Overall Solution Strategy

To this end, all the components of the strategy are specified. The problem to be solved can be formulated as: "given the nodal position, the velocity, the pressure, and temperature in both domains and exclusively in the air domain, the mass fraction of fuel $Y_F$ and the oxygen $Y_O$ as well as the incident radiation G at time $t^n$ find these variables at $t^{n+1}$". The overall solution strategy is summarized in Algorithm 1.

## 4. Model and Numerical Computation

The model was implemented in Kratos Multi-Physics code [28]. This section is organized as follow: First, the specimen as well as the results of its experimental material characterization required for the numerical tool are presented. Moreover, the results of the UL 94 test together with the measurement of the temperature are presented. Next, details of the computational domain are presented. In addition to this, the initial and boundary conditions including the input parameters for both solvers are introduced. This section concludes with the simulation of the UL 94 test and a comparison of its results versus the experimental ones.

---

**Algorithm 1:** Solution algorithm for the simulation of the UL 94 test.

---

1 **for** $t = t^{n+1}$ **do**

    in the air domain

      • Fix on the interface the velocity and temperature (Equation (2)) following [19];

      • Solve:

          – RTE eq. (Equation (16));

          – Energy eq. (Equation (17));

          – Navier–Stokes eq. (Equations (19) and (20));

    in the polymer domain

      • Prescribe the normal heat flux $q_R$ provided by the air at the surface;

      • Solve:

          – Energy eq. (Equation (15));

          – Navier–Stokes eq. (Equations (13) and (14)).

2 **end**

---

### 4.1. Materials, Experimental Methods, and Input Parameters

#### 4.1.1. Material

The mixing of MDH (Sigma Aldrich) at 30 wt.% and polypropylene (ISPLEN PP-045 G1E-PP YPF, $\rho = 905$ kg/m$^3$, $C = 1.8$–$2.0$ kJ/Kg °C, $\kappa = 0.1$–$0.22$ Wm$^{-1}$K$^{-1}$) was conducted by first melt-blending in a high-torque microcompounder (Xplore MC 15) at 100 rpm, 190 °C for 3 min in each batch. Afterwards, the compounding material was heat molded using a platen hot-press (LabPro 400, Fontijne Presses). All of the samples were pressed at 2 MPa for 10 min at 190 °C.

#### 4.1.2. Thermogravimetric Analysis

In order to estimate the kinetics parameters (Ea and A, activation energy and Arrhenius coefficient, respectively), the pyrolysis process was carried out in a thermogravimetric analyzer (TGA Q50, TA Instruments) under nitrogen atmosphere. For further details on the procedure, the reader is referred to [29,30]. There exist two main groups that can be used in the analysis of the kinetics in non-isothermal and solid-state mode obtained from the TGA (fitting and free models) [31]. Free models are more often implemented than free ones, as the latter possess more inherent complications with regard to the selection of the kinetic model [32] than those of the former. Free models estimate activation energy and the Arrhenius coefficient by using different heating rates and obtaining a range of activation energy as a function of different conversion values. Flynn–Wall–Ozawa and Kissinger methods are the non-isothermal free models selected to calculate both kinetic parameters.

In the case of the Kissinger model, the values calculated are 181.9 kJ/mol and $1.63 \times 1014$ min$^{-1}$ for activation energy and the Arrhenius coefficient, respectively, while for the FWO method, they are 84.75 kJ/mol and 28,883.43 min$^{-1}$ for activation energy and the Arrhenius coefficient, respectively.

#### 4.1.3. Rheological Behavior

To obtain experimental values of viscosity MDH30/PP samples, a rheometer (AR200EX, TA Instruments) was used. The dimensions of the samples were 25 mm in diameter and 0.5 mm in thickness. The experiments were conducted according to the evolution of the viscosity of the material in the dynamic temperature step mode ranging from 453.15 to 533.15 K. A low value of shear rate (0.1 rad/s) was chosen for the experimental tests since the process of dripping occurs when this value is low [6]. Figure 2 exhibits the values of viscosity depending on the temperature.

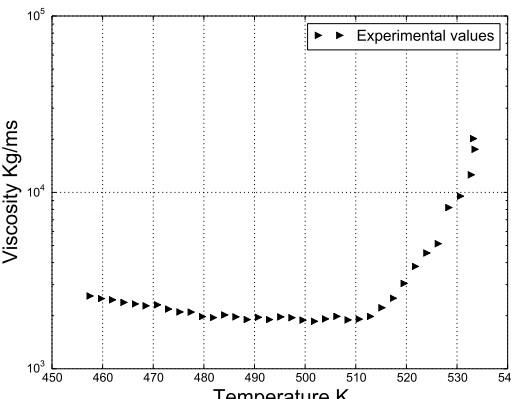

**Figure 2.** Viscosity of MDH30/PP in the dynamic temperature step mode.

### 4.1.4. UL 94 Test

The UL 94 test [33] was performed in a burning chamber (UL 94, FTT, UK) Technology, United Kingdom). The dimensions of the specimens were $127 \times 12.7 \times 3.2$ mm$^3$, and each of them was equipped with three type-K thermocouples (Inconel 600, diam. 0.75 mm, long. 250 mm, Tmax = 800 °C) located in different sections of the specimens (Figure 3). The flame was inflicted at the bottom of the specimen for 10 s. Once this time passed, the flame was taken out, and the fire did not stop spreading along the sample, as shows in Figure 4. As a consequence of this behavior during the test, the material was classified as "no rating" according to [5].

### 4.2. Numerical Setup

The geometry and dimensions of the computational domain are presented in Figure 5a. The domain of the polymer and air are discretized by a non-structured tetrahedra mesh of 100,902 and 286,850 elements, respectively (Figure 5b). Note that only the fourth part of the whole problem was considered when assuming symmetry conditions. This assumption is based on the fact that although the flame was applied with a Bunsen burner at 45 degrees, it was always in full contact with the bottom of the polymer rather than with the vertical sides. No vertical side was ever directly exposed to the flame. Thus, in this example, the symmetry was never lost, and, therefore, the heat provided by the flame was maximum in the lower part of the polymer [6,7,34]. In fact, previous studies have explored the same idea independently of the size of the domain for simulation (one fourth or full domain) [8,34] and with consideration of the fact that the heat of the specimen has a varied height.

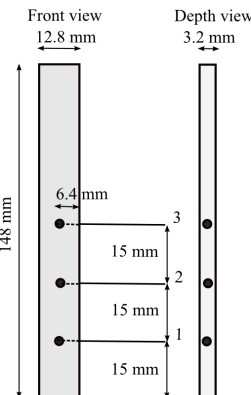

**Figure 3.** Position of the thermocouples .

Regarding the boundary conditions (Figure 5a), for both domains over the symmetric faces, we set the normal velocity component and the normal gradients of all flow variables (including temperature) to zero. The clamping of the polymer and the Bunsen were

modelled by fixing all of the velocity components to zero and applying a face heat flux, respectively. The latter was set to 85 [kW/m$^2$] during the first 10 s following the criteria adopted in [8]. On the other hand, a constant uniform velocity was prescribed in the inlet and a pressure value at the outlet of the air domain. The constant velocity boundary condition is indeed an approximation. Considering variable velocity, the boundary condition is possible; however, such a boundary condition leads to the divergence of the numerical solver due to the appearance of velocities in some parts of the inlet boundary pointing in the opposite direction (downwards).

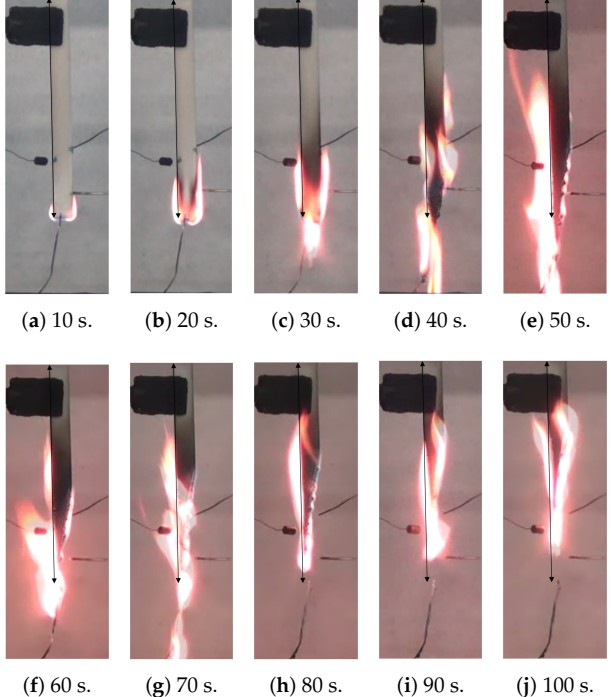

| (**a**) 10 s. | (**b**) 20 s. | (**c**) 30 s. | (**d**) 40 s. | (**e**) 50 s. |

| (**f**) 60 s. | (**g**) 70 s. | (**h**) 80 s. | (**i**) 90 s. | (**j**) 100 s. |

**Figure 4.** Images of the UL 94 test at different time steps.

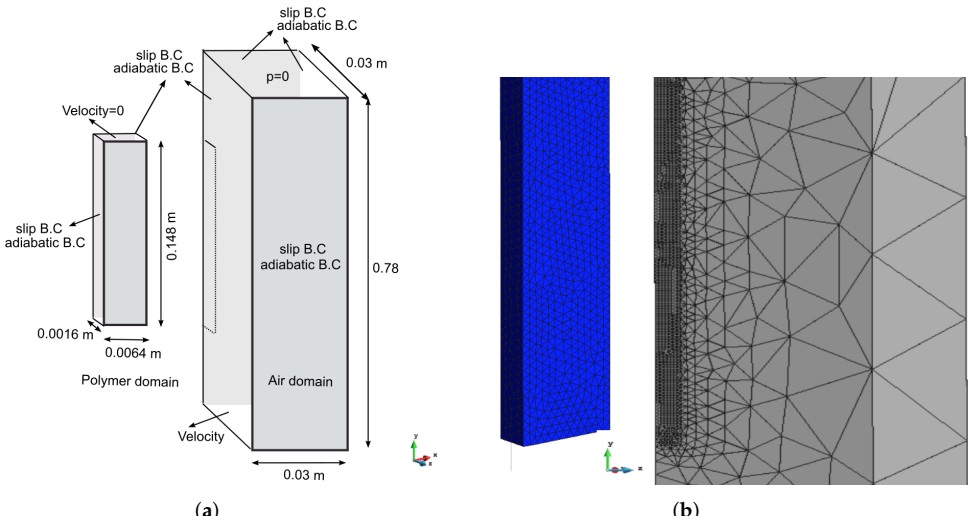

**Figure 5.** Problem definition. (**a**) Computational domains with boundary conditions. (**b**) Meshes of the polymer and the air .

On the other hand, the initial conditions consisted of initializing the gas temperature, oxygen, and fuel mass fractions to 298 K, 0.23, and 1, respectively. Once the surface of the polymer reached the ignition temperature, $Y_O$ was fixed to 0 and $Y_F$ to 1.

The input parameters for both solvers are summarize in Table 1.

**Table 1.** Input parameters for the PFEM and Eulerian solvers.

| Parameter | Polymer | Air |
|---|---|---|
| Density | 905 [kg/m$^3$] | see Section 2 |
| Viscosity | ¯(T) | $1 \times 10^{-5}$ [m$^2$/s] |
| Specific heat capacity | 1900.0 [J/KgK] | 1310.0 [J/KgK] |
| Thermal conductivity | 0.16 [W/mK] | 0.0131 [W/mK] |
| Emissivity | 1.0 | – |
| absorption coefficient | – | 1000 [m$^{-1}$] |
| Stefan–Boltzmann constant | – | $5.67 \times 10^{-8}$ [W/m$^2$K$^4$] |
| Arrhenius coefficient | $1.63 \times 10^{14}$ [min$^{-1}$] | – |
| Activation energy | 181.9 [KJ/mol] | – |
| Enthalpy of vaporization | $8 \times 10^5$ [W/m$^2$K] [12] | – |
| B$_c$ | – | $5.96 \times 10^9$ [m$^3$/Kgs] |
| T$_a$ | – | 10,700 [K] |
| C | – | $2.601 \times 10^4$ [Kj/Kg] |

The expression of the adjusted curve corresponding to the experimental viscosity values presented in Figure 2 is given in Algorithm 2 . Note that the viscosity is reported to be 453.15 to 533.15 K. However, the registered temperatures of the polymer during the flame test are clearly higher (Figure 6).

As viscosity is the key parameter for modeling the dripping behavior [6,7], in this work, it was estimated on the basis of the viscosity curve provided by NIST [12] and the images of the experiment for those ranges where the curve was not available. These two sources allowed us to estimate the viscosity–temperature curve as follows: (i) if temperature < 453.15 K, viscosity is directly approximated using the NIST curve [12]; (ii) if temperature > 533.15 K and temperature < 724 K, viscosity is set to 20,000 [m$^2$/s] (last known value of the rheological study) since the polymer does not considerably move according to the experiment ); (iii) if temperatures > 725 K, three ways to decrease the viscosity are proposed following the pattern showed by the NIST curve. It should be noted that these curves are from hereon referred to as curve 1, curve 2, and curve 3. These are presented in the Appendix A and plotted in Figure 7. Figure 7 also shows the viscosity–temperature dependency of the pure PP in [12].

Although the evolution of the viscosity is proposed for our case, we do not have a sufficient amount of information regarding the variation of the other parameters. However, in recent works, constant values have been used [6,7,9,12,14–17] in absence of better information. This is supported by the studies based on simulation results conducted by Stoliarov et al. [35], who concluded that parameters such as density, thermal conductivity, and heat capacity are of little importance to the burning behaviors of polymers in comparison with other parameters affecting combustion.

---

**Algorithm 2:** Definition of the experimental viscosity curve.

1  if(T > 453.15 and T <= 515.88):

2    mu = $1.25 \times 10^4 \times e^{(-3.4 \times 10^{-3} \times T)}$

3  else if(T > 515.88 and T <= 533.15):

4    mu = $1.10 \times 10^{-25} \times e^{(1.26 \times 10^{-1} \times T)}$

---

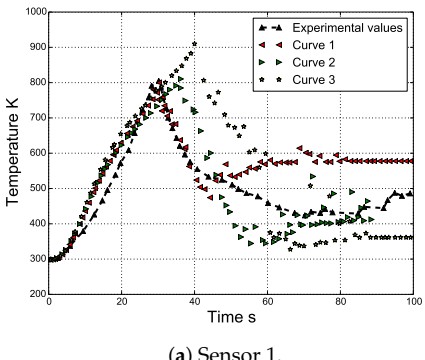

(**a**) Sensor 1.

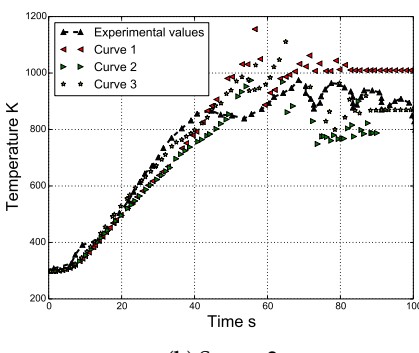

(**b**) Sensor 2.

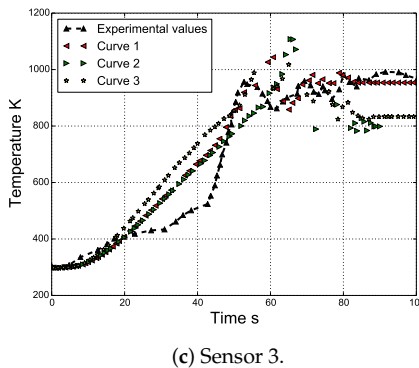

(**c**) Sensor 3.

**Figure 6.** Temperature evolution for sensors 1, 2, and 3.

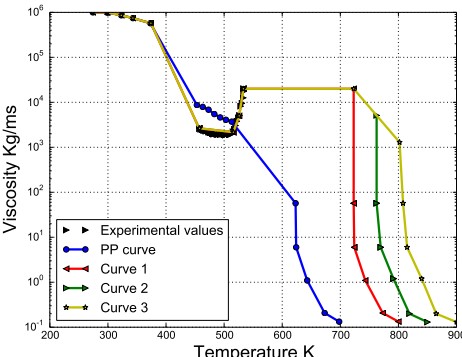

**Figure 7.** Viscosity curve.

### 4.3. Numerical Results

Figures 8–10 compare the experimental versus numerical results corresponding to the proposed curves. In these figures, the initial length of the specimen is indicated by a continuous line. Note that the polymer specimen (main body) loses mass due to two mechanisms: (i) gasification and (ii) melting and dripping. Both phenomena are strongly interrelated due to the fact that the dripping modifies the temperature of the surface and, therefore, the mass loss rate due to gasification. Figure 11a shows the volume loss curves for the three curves.

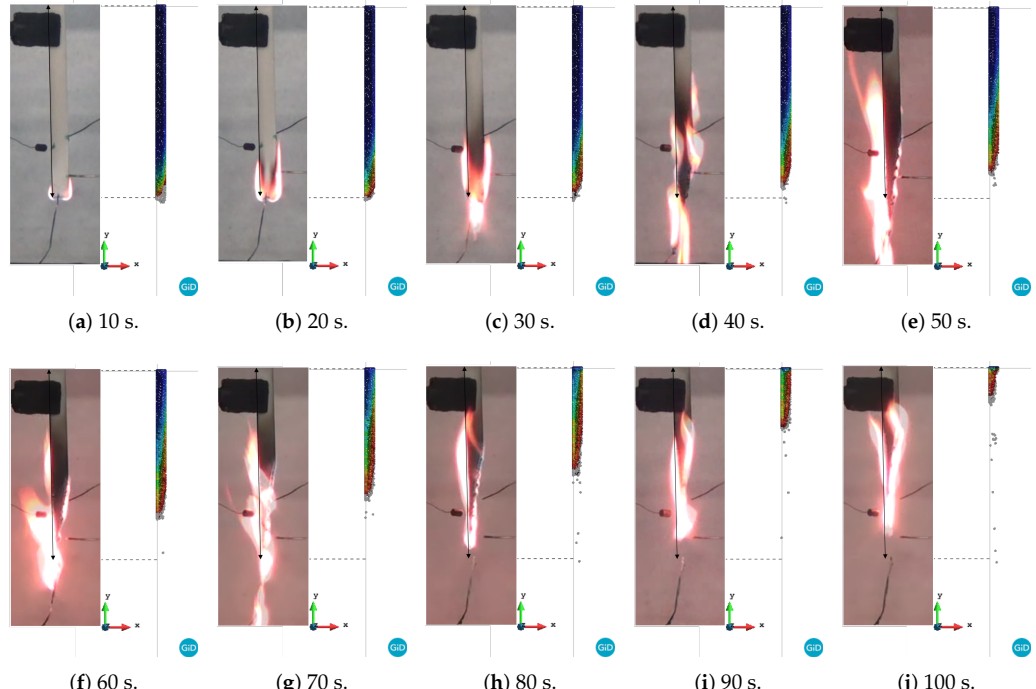

| (a) 10 s. | (b) 20 s. | (c) 30 s. | (d) 40 s. | (e) 50 s. |
|---|---|---|---|---|

| (f) 60 s. | (g) 70 s. | (h) 80 s. | (i) 90 s. | (j) 100 s. |
|---|---|---|---|---|

**Figure 8.** Comparison between experimental versus PFEM results for curve 1: blue and red correspond to 298 and 1000 K, respectively.

A quick inspection of the figures reveals good agreement in the shapes between the experimental and numerical results for all of the curves up to 20 s. This is due to the fact that the three cases have the same viscosity until 723.15 K. As time progresses, the difference between the three cases becomes more evident. For instance, case 1 begins to reduce its volume due to melting and dripping before the experiment does. On the other hand, case 3 shows behavior dominated more by melting than by dripping. In fact, the length and width of the model are larger than those of the experimental model until 60 s (Figure 11b). Finally, case 2, whose curve is between case 1 and 3, shows better agreement with the experimental results in terms of shapes of the specimen than that of the other cases. However, none of these cases is able to capture the thread of molten material that appears after 70 s. This behavior is probably linked to the surface tension effects, which are not considered in this work. Moreover, a higher temperature at the edges due to the edge effect was observed [36,37]. In addition, according to Figures 8–10, the sample did not stop the fire from spreading as in the case of the UL 94 test.

The results shown in Figure 12a present the computed temperature distribution across four horizontal cut planes at different heights of the air domain for curve 2. As time progresses, the heat zone develops in the air domain, providing heat feedback to the specimen. As a consequence, thee temperature in the polymer evolves, leading to a reduction in the viscosity value corresponding to curve 2. Due to this viscosity change, the polymer starts to melt and drip. The aforementioned phenomena together with the gasification are responsible for the volume reduction in the sample. As can be observed

in Figure 12b, the maximum concentration of fuel appears in the polymer–air interface when the polymer reaches the gasification temperature due surface exposure to a heat flux. Finally, this fuel mixes with the oxygen and reacts, thereby producing the combustion.

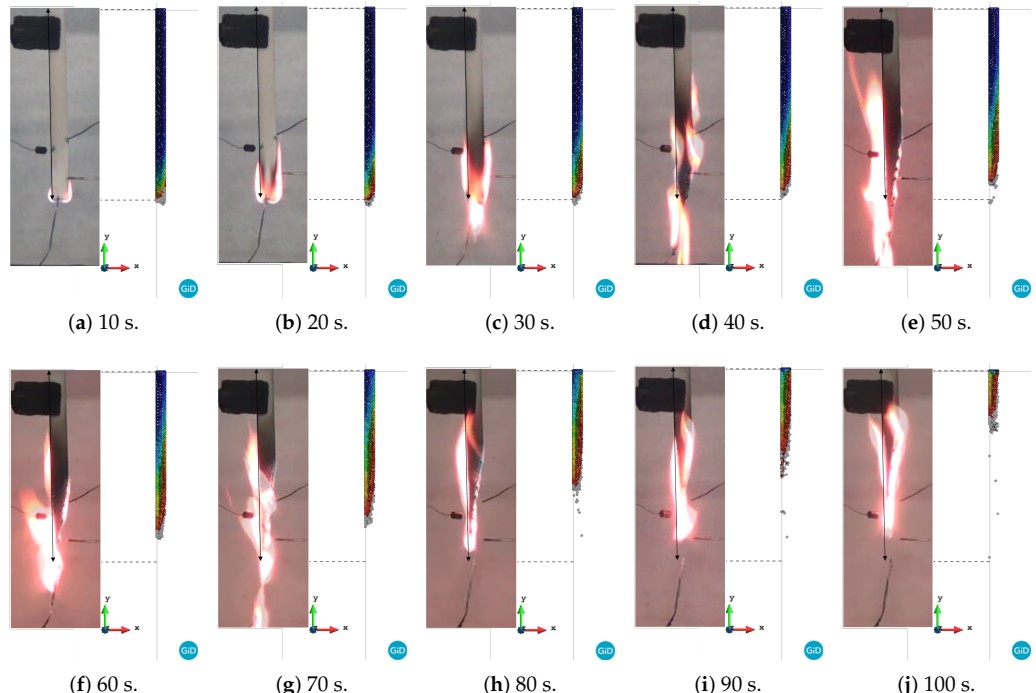

**Figure 9.** Comparison between experimental versus PFEM results for curve 2: blue and red correspond to 298 and 1000 K, respectively.

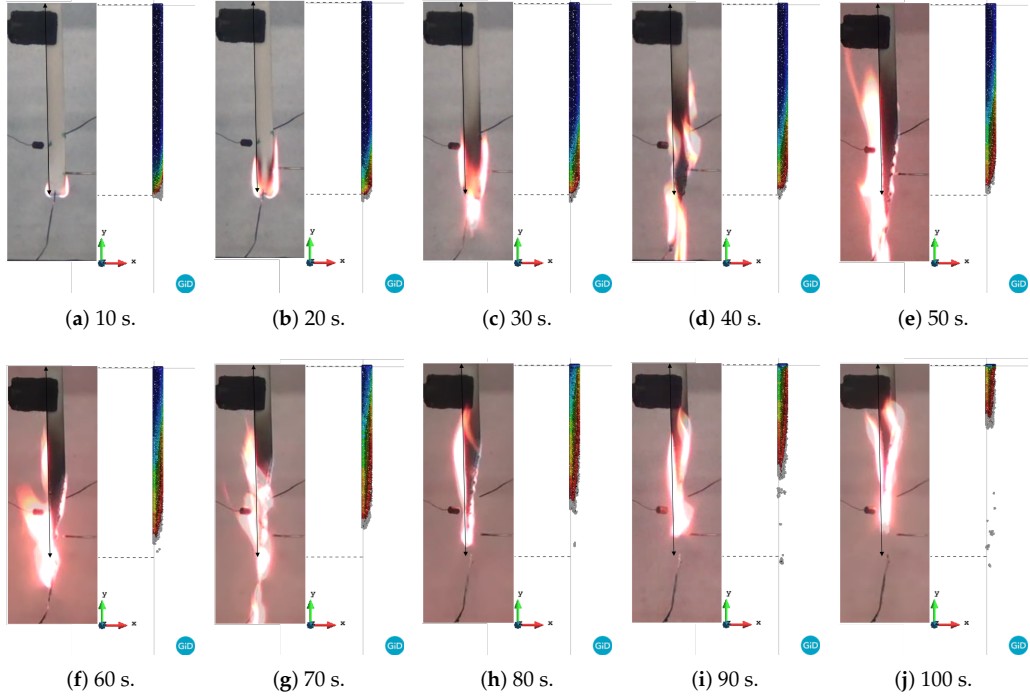

**Figure 10.** Comparison between experimental versus PFEM results for curve 3: blue and red correspond to 298 and 1000 K, respectively.

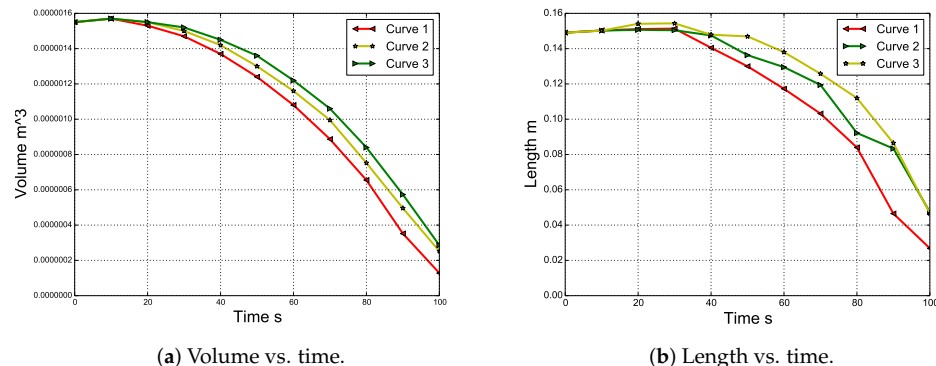

(**a**) Volume vs. time.　　　　　　　　(**b**) Length vs. time.

**Figure 11.** Evolution of the length and volume simulated with PFEM.

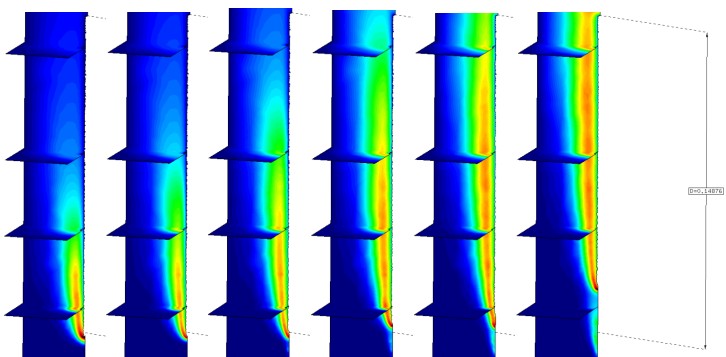

(**a**) Temperature distribution: blue and red correspond to 298 and 1400 K, respectively.

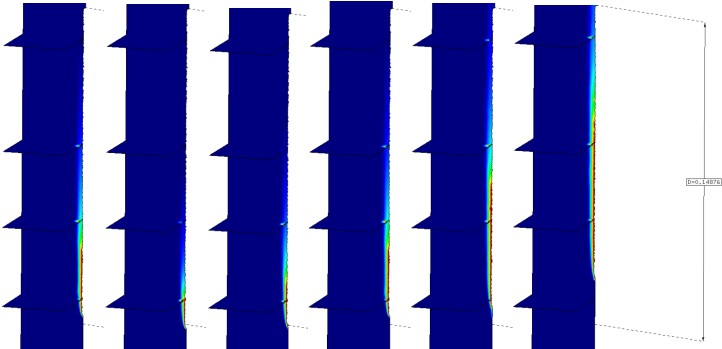

(**b**) Fuel distribution: blue and red correspond to 0 and 1, respectively.

**Figure 12.** Evolution of the temperature and fuel distribution in the air domain at 10, 20, 30, 40, 50, and 60 s.

Figure 6 compares the experimental and numerical temperatures at different times for thermocouples 1, 2, and 3. The latter corresponds to the finite element, where the thermocouple can be observed, as the size of the mesh coincides with the diameter of the thermocouple. In all cases, the numerical strategy captures considerably well the rising branch of the curve, which corresponds to when the thermocouples are inside the polymer. Note that its extension depends on the evolution of the shape of the polymer, which is governed by the melting and dripping of the material. Moreover, we note that as we move from sensor 1 in the direction of sensor 3, the discrepancies between the experimental and the numerical values are more evident. This difference could originate due to the result of some complex phenomena not considered in this work, such as bubbling, micro-explosions, char formation, and complex time-dependent decomposition.

Subsequently, the remaining curve represents the measurements of the temperature corresponding to the flame or melted material passing through the thermocouples. It is important to remark that all of the thermocouples for an unknown reason move away from their original positions. Note that this is evident for thermocouple 1, where the recorded temperatures decrease. In order to consider this effect, an approximated trajectory was taken from the images. Although this procedure is limited by the quality of images, the numerical results corresponding to the trajectory of the thermocouples follow the same trend as that observed in the experimental results. In particular, the results corresponding to curve 2 are those that best fit the experimental results.

One can conclude that knowing the change in viscosity of PP due to the addition of MDH is crucial to achieving good agreement between the numerical results and the evolution of the temperature measurements given by the three thermocouples positioned inside the polymer. Accounting for the viscosity change is also crucial to the prediction of the evolution of the polymer shape.

Although the correlation observed is not perfect, given the complexity of the set-up, we consider the agreement between the experiments and the simulation results satisfactory.

## 5. Conclusions

In this paper, the numerical tool developed by the authors was applied for modelling the dripping and melting behavior of a blend of polypropylene mixed with a commercial halogen-free fire retardant (30 wt.% MDH, magnesium hydroxide) in a UL 94 fire scenario set-up coupled with temperature monitoring.

The input parameters required for the model were obtained from different literature sources/diverse manufacturers and previously performed experimental studies.

The numerical results reveal that a well-adjusted viscosity of the material is crucial to achieving good agreement in the evolution of the temperature measurements given by the three thermocouples positioned inside the polymer as well as in the evolution of the shape of the polymer. Thus, the effect of MDH on the viscosity curve of pure PP and its important role in PP's behavior in fire situations were examined. This allowed for us to improve our understanding of the complex behavior of polymeric materials during fires.

To further improve the model, examination of polymer material properties as a function of temperature and a better characterization of the chemical reaction in the air are needed. However, the inclusion of temperature-dependent parameters is trivial in our model. Therefore, in the case of obtaining these dependencies from the experiments, they can be immediately integrated into the model.

**Author Contributions:** J.M.: software, validation, visualization, and writing—original draft. J.d.l.V.: formal analysis, investigation, and writing—original draft. D.-Y.W.: supervision, methodology, and writing—review and editing. E.O.: writing—review and editing. All authors have read and agreed to the published version of the manuscript.

**Funding:** This document is the results of the research project funded by the COMETAD project of the National RTD Plan (ref. MAT2014-60435-C2-1-R) from the *Ministerio de Economía y Competitividad* of Spain. The authors Julio Marti and Eugenio Oñate acknowledges financial support from the Ministerio de Ciencia, Innovacion e Universidades of Spain via the Severo Ochoa Programme for Centres of Excellence in RD (referece: CEX2018-000797-S)

**Institutional Review Board Statement:** Not applicable.

**Informed Consent Statement:** Not applicable.

**Conflicts of Interest:** The authors declare no conflict of interest.

## Appendix A

The viscosity curves corresponding to the images in Figure 7 are as follows

---

**Algorithm A1:** Definition of the curve 1.

---
1 If(T <= 373.0):
2    mu=$5.72 \times 10^6 \times e^{(-6 \times 10^{-3} \times T)}$
3 else if(T > 373.0 and T <= 453.15):
4    mu = $1.95 \times 10^{16} \times e^{(-6.48 \times 10^{-2} \times T)}$
5 else if(T > 453.15 and T <= 515.88):
6    mu = $1.25 \times 10^4 \times e^{(-3.4 \times 10^{-3} \times T)}$
7 else if(T > 515.88 and T <= 533.15):
8    mu = $1.10 \times 10^{-25} \times e^{(1.26 \times 10^{-1} \times T)}$
9 else if(T > 533.15000001 and T <= 724.0):
10    mu = 20000.0
11 else
12    mu = $1.83 \times 10^{16} \times e^{(-4.98 \times 10^{-2} \times T)}$

---

**Algorithm A2:** Definition of the curve 2.

---
1 if(T <= 533.15):
2    Definition Curve 1
3 else if(T > 533.15000001 and T <= 723.0):
4    mu = 20000.0
5 else if(T > 723.0 and T <= 762.49):
6    mu = $1.38 \times 10^{15} \times e^{(-0.035 \times T)}$
7 else if(T > 762.49 and T <= 769.5):
8    mu = $1.70 \times 10^{266} \times e^{(-7.94 \times 10^{-0.001} \times T)}$
9 else if(T >769.5 and T <= 791.5):
10    mu = $1.49 \times 10^{25} \times e^{(-7.30 \times 10^{-2} \times T)}$
11 else if(T >791.5 and T <= 819.0):
12    mu = $2.99 \times 10^{22} \times e^{(-6.52 \times 10^{-2} \times T)}$
13 else if(T >819.0 and T <= 850.0):
14    mu = $1.75 \times 10^4 \times e^{(-1.39 \times 10^{-2} \times T)}$
15 else
16    mu = 0.13

---

**Algorithm A3:** Definition of the curve 3.

---
1 if(T <= 533.15):
2    Definition Curve 1
3 else if (T > 533.15000001 and T <= 723.0):
4    mu = 20000.0
5 else if(T > 723.0 and T <= 802.0):
6    mu = $2.0 \times 10^{15} \times e^{(-0.035 \times T)}$
7 else if (T > 802.0 and T <= 815.0):
8    mu = $2.0 \times 10^{146} \times e^{(-0.411 \times T)}$
9 else if (T > 815.0 and T <= 900.0):
10    mu = $9.0 \times 10^{16} \times e^{(-0.046 \times T)}$
11 else
12    mu = 0.13

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
