# Peer review of "Numerical Simulation of Flame Retardant Polymers Using a Combined Eulerian–Lagrangian Finite Element Formulation"

_applsci, doi:10.3390/app11135952_

Round 1
Reviewer 1 Report
This paper is interesting but needs some revision before publication.
Section 4 is a mess. Please carefully divide 1) experiments 2) simulation. Those seems mixed up in current section 4. It would be better if section 4 could be divided section 4 and 5.
Figures 7-9, simulation is too small to compare with experimental results. Please make them bigger for readability.
Author Response
Dear Reviewer,
thank you very much for your feedback and valuable comments. Please find below the detailed response to the queries.
1-Section 4 is a mess. Please carefully divide 1) experiments 2) simulation. Those seems mixed up in current section 4. It would be better if section 4 could be divided section 4 and 5.
Thanks for your comment. In order to improve the organization of section 4 (“Model and numerical”), a new subsection called “Material, experimental methods and input parameters” was created to contain all the experimental data. The changes are in red color.
2-Figures 7-9, simulation is too small to compare with experimental results. Please make them bigger for readability.
The size of the figures was chosen in to order to have in one page all the figures together. Otherwise the figures will occupy several pages. I think that this a task that the editor could do.
Reviewer 2 Report
- The layout of the manuscript must be carefully improved as, at least in this pdf version submitted for the review process, some figures are inserted in between the references (pages 14,16); inconsistent use of spaces between the text and reference numbers (e.g. lines 71, 103, 104, etc.) or in such cases as in lines 94, 106, etc.; inconsistent use of decimal separators (e.g. lines 179, 181).
- Excerpts of codes (lines 243-260; 268-207) should be considered to be presented in appendixes. Also, it would be more appropriate to present them as block diagrams.
- A comparison of the obtained modeling results to results obtained by some other methods would be useful and interesting in order to highlight the advantages of the numerical tool developed by the authors.
Author Response
Dear Reviewer,
thank you very much for your feedback and valuable comments. Please find below the detailed response to the queries.
1-The layout of the manuscript must be carefully improved as, at least in this pdf version submitted for the review process, some figures are inserted in between the references (pages 14,16);
It was corrected.
2-inconsistent use of spaces between the text and reference numbers (e.g. lines 71, 103, 104, etc.) or in such cases as in lines 94, 106, etc.;
It was corrected.
3-inconsistent use of decimal separators (e.g. lines 179, 181).
It was corrected.
4-Excerpts of codes (lines 243-260; 268-207) should be considered to be presented in appendixes. Also, it would be more appropriate to present them as block diagrams.
Thanks for your comment. The viscosity curves have been moved to an appendix.
5-A comparison of the obtained modeling results to results obtained by some other methods would be useful and interesting in order to highlight the advantages of the numerical tool developed by the authors.
We agree with the reviewer, however there is no code available capable of doing this type of simulation. The numerical results are compared with the experimental ones.